# Improved 1-km-Resolution Hourly Estimates of Aerosol Optical Depth Using Conditional Generative Adversarial Networks

**Luo Zhang [1,2], Peng Liu [1,\*], Lizhe Wang [3], Jianbo Liu[1], Bingze Song [1], Yuwei Zhang [1], Guojin He [1] and Hui Zhang [4]**

1   Aerospace Information Research Institute, Chinese Academy of Sciences, No. 9 Dengzhuang South Road, Haidian District, Beijing 100094, China; zhangluo@radi.ac.cn (L.Z.); liujb@aircas.ac.cn (J.L.); songbingze19@mails.ucas.ac.cn (B.S.); zhangyuwei20@mails.ucas.ac.cn (Y.Z.); hegj@aircas.ac.cn (G.H.)
2   School of Electronic, Electrical and Communication Engineering, University of Chinese Academy of Sciences, Beijing 100094, China
3   School of Computer Science, China University of Geosciences, Wuhan 430074, China; Lizhe.Wang@gmail.com
4   Beijing Advanced Innovation Center for Big Data-Based Precision Medicine, School of Engineering Medicine, Beihang University, Beijing 100191, China; hui.zhang@buaa.edu.cn
\*   Correspondence: liupeng@radi.ac.cn

**Abstract:** Aerosol Optical Depth (AOD) is a crucial parameter for various environmental and climate studies. Merging multi-sensor AOD products is an effective way to produce AOD products with more spatiotemporal integrity and accuracy. This study proposed a conditional generative adversarial network architecture (AeroCGAN) to improve the estimation of AOD. It first adopted MODIS Multiple Angle Implication of Atmospheric Correction (MAIAC) AOD data to training the initial model, and then transferred the trained model to Himawari data and obtained the estimation of 1-km-resolution, hourly Himawari AOD products. Specifically, the generator adopted an encoder–decoder network for preliminary resolution enhancement. In addition, a three-dimensional convolutional neural network (3D-CNN) was used for environment features extraction and connected to a residual network for improving accuracy. Meanwhile, the sampled data and environment data were designed as conditions of the generator. The spatial distribution feature comparison and quantitative evaluation over an area of the North China Plain during the year 2017 have shown that this approach can better model the distribution of spatial features of AOD data and improve the accuracy of estimation with the help of local environment patterns.

**Keywords:** aerosol; conditional generative adversarial network; spatio-temporal estimation





## 1. Introduction

Spatio-temporal completed and accurate AOD products are fundamental data to various aerosol-related studies [1]. AOD can be obtained from ground-based measurements, satellite retrieves, and model simulations; each has its advantages and disadvantages. The AERONET (AErosol RObotic NETwork) can provide accurate globally distributed observations of spectral AOD but limited spatial coverage [2]. Model-simulated AOD such as the Global 3D Atmospheric Chemical Transport model (GEOS-Chem) [3] driven by the Goddard Earth Observing System (GEOS) and the Weather Research and Forecasting model coupled to Chemistry (WRF-Chem) [4] could generate aerosol profiles and column concentrations with a high temporal resolution, but the spatial resolution and accuracy are still limited. In contrast, satellite AOD retrievals, such as the Moderate Resolution Imaging Spectro-Radiometer (MODIS) [5] and the Advanced Himawari Imager (AHI) [6], are widely used because of their extensive spatial coverage. However, due to the limited swath width, the influences of cloud cover, and the inherent theoretical limitation of AOD retrieval algorithms, the AOD products from a single satellite sensor are still not enough in accuracy and spatiotemporal completeness. The complementary characteristics of various AOD products from different satellites provided possibilities to estimate the spatiotemporal

distributions of AOD. Therefore, taking advantage of the strengths of the different satellite AOD products, we could generate high spatiotemporal AOD products with more integrity and quality [7,8].

Usually, most methods for merging multi-sensor AOD products are mainly spatial and temporal interpolation, which utilize the neighborhood pixel values of an AOD product to fill in the missing values of other types of AOD products at the same locations [9]. For instance, according to the relationship of group AOD pixel values at the same geographic locations from different satellite sensors, researchers developed polynomial regression models [10], maximum likelihood estimation models [11], least square estimation models [12], optimal interpolation [13], and some simplified merge schemes [14]. Additionally, there are some geostatistical methods, including the universal kriging method [15], the geostatistical inverse modeling [16], and the spatial statistical data fusion [17]. These interpolation methods mainly focus on improving spatial coverage. Moreover, they usually lead to smooth diffusion and increase the uncertainty of AOD products.

In recent years, some deep learning and data-driven methods achieved promising results in remote sensing data processing [18–20], and they also have received more attention in some aerosol-related studies. For example, Tang et al. [21] developed a spatiotemporal fusion framework based on a Bayesian maximum entropy, which is used for the MODIS and the Sea-viewing Wide Field-of-view Sensor AOD products. Zhao et al. [22] utilized statistical machine learning algorithms to train the model for estimating AOD values, which seeks the relationships between AOD retrievals from satellites and other factors (e.g., meteorological parameters). Chen et al. [23] proposed an artificial neural network for aerosol retrieval, which is implemented for joint retrieval of MODIS AOD and fine mode fraction. These methods show some of the advantages of machine learning in merging multi-sensor AOD products. However, very few studies have focused on the improvement of spatial resolution of AOD by using the deep learning method.

According to the above analysis, this study proposes a conditional generative adversarial networks-based architecture (AeroCGAN) and training strategy to estimate high spatiotemporal resolution AOD. The MODIS Multiple Angle Implication of Atmospheric Correction (MAIAC) algorithm utilizes multi-angle information from time series of MODIS observation for up to 16 days for a given pixel at the resolution of 1 km. It enables retrieval of aerosol loading at high resolution of 1 km, providing an excellent opportunity for aerosol research at finer spatial scales [24]. Current C6 MODIS MAIAC aerosol products have been comprehensively evaluated [5,25]. Since the MAIAC algorithm has striking advantages in cloud and snow/ice screening, spatial coverage, and pixel resolution, the completeness of the MAIAC product may be higher than other products. The statistics with the ground-truth AERONET data showed that 69.84% of retrievals fall within the expected error envelope and the correlation coefficient is greater than 0.9, indicating a good accuracy for MAIAC products in China. Moreover, MAIAC AOD values exhibit higher accuracy in Beijing and Xianghe, the main study areas of this research. Therefore, we first adopted MODIS MAIAC AOD data to train the model and then transferred the trained model to Himawari data and finally obtained 1-km-resolution, hourly estimates of the Himawari AOD product. Specifically, for the generator, it adopted an encoder–decoder network for preliminary resolution enhancement, then used a three-dimensional convolutional neural network (3D-CNN) [26] for environment features' extraction, and finally added the features to a residual network [27] for improving the accuracy. Meanwhile, the sampled data and environment data were designed as conditions of the generator. The spatial visual comparison and quantitative evaluation have both shown that this approach achieved competitive performances.

Specifically, the main contributions of this paper include:

- Taking advantage of the high spatial resolution of MODIS MAIAC AOD products and high temporal resolution of Himawari AOD products, a data-driven method is proposed to improve the spatio-temporal resolution of AOD. It uses MAIAC AOD

as training data to capture the complex spatial patterns and perform estimation to Himawari AOD based on this learned knowledge.

- According to the features of AOD data and the correlation between auxiliary data (e.g., meteorological, land-related data are described in Section 2.3) and AOD, the proposed model AeroCGAN constructs two conditions: the sampled data as a spatial condition for generating reasonable spatial distribution; and the environment features extracted from auxiliary data as an environmental condition for improving the accuracy and producing more realistic details.
- The model can effectively capture complex spatial patterns and preprocess the data with an active window selection strategy. In this way, the model could increase spatial coverage and generate high spatial resolution AOD with more realistic details, which is on a reasonable spatial scale.

The rest of this paper is organized as follows: Section 2 first explains the fusion framework for spatiotemporal AOD, then describes the detailed structure of the proposed network, including the conditions, generator, discriminator, and loss function design. Section 3 introduces the study area and experimental dataset, the evaluation metrics, and presents the experimental results of different methods. In addition, the comparison, validation, and analysis between the original Himawari 5 km AOD and generated high Himawari 1 km AOD are also performed. Finally, the overall conclusion of this paper is summarized in Section 4.

## 2. Materials and Methods

### 2.1. The Fusion Framework for Spatiotemporal AOD

This task of estimating high spatiotemporal AOD is similar to spatiotemporal fusion and super-resolution (resolution enhancement) in some fields. Nevertheless, the AOD data are quite different from the natural or optical remote sensing images. Specifically, we considered the following differences between common optical images and AOD data as follows.

- Spatiotemporal Difference: The surface reflectance shows temporally slow and spatially high variations, whereas the aerosol loading changes very fast over time and varies only on a limited space scale. Different from natural images, the variations in remote sensing images are mainly caused by phenology, seasons, disaster, or human activities. Most of the land surface changes in multi-temporal optical remote sensing can be regarded as a relatively independent slow feature for analysis [28]. In contrast, AOD is a physical quantity that characterizes the degree of atmospheric turbidity, which has different spatiotemporal heterogeneity and dramatic variability; meanwhile, it has strong correlation with other atmospheric or geographic environmental information data [29];
- Spectral Difference: Usually, natural images contain three bands of red, green, and blue. Optical remote sensing images usually have multiple bands which can provide more information for the analysis of characteristics. However, AOD data have its own physical meaning that is different from multi-spectral or hyperspectral images;
- Feature Difference: The features in natural images usually have strong logical correlation. In addition, high- and low-resolution natural images are basically coherent in visual structure information. Optical remote sensing has complex feature types and rich textural features, and their features have lower logical correlation. However, features of AOD images tend to be monotonous and poor, so that the complexity of the spatiotemporal heterogeneity makes its estimation and validation more difficult.

Because of the above differences, super-resolution methods for natural images or spatial-temporal fusion algorithms for optical remote sensing images are not entirely suitable for AOD resolution enhancement. In addition, there is information complementary of multi-sensor and auxiliary environmental data, which could provide more different

supplements for AOD estimates. For example, the MODIS MAIAC AOD products have a high spatial resolution, the Himawari AOD products have a high temporal resolution, and the auxiliary data (e.g., meteorological, land-related data) have a correlation with AOD [22,30]. These data are helpful for estimating high spatiotemporal resolution AOD. In this paper, we adapted the generative and adversarial architectures to fuse the multi-source information and improve the resolution of Himawari AOD.

The flowchart of the proposed method is shown in Figure 1. For clarity, Table 1 lists the notations in the following sections and Figure 1, where $e_m$ and $e_h$ are the environment features that match MAIAC and Himawari, respectively, extracted from auxiliary data by 3D-CNN networks. Overall, the framework mainly contains two stages: (a) training generator $G$ with MAIAC AOD data, and (b) applying the trained generator $G$ to Himawari AOD data. Considering the strong temporal and spatial correlation between the MODIS MAIAC and the Himawari AOD products in the studied region, the trained model with MAIAC AOD could be applied to Himawari AOD. Therefore, during stage (a), we aim to train the generator $G$ with those MODIS MAIAC AOD. $M_{HR}$, $M'_{LR}$, and $M'_{HR}$ constitute the corresponding high- and low-resolution data, and $e_m$ will help to improve the model accuracy. During stage (b), we apply the trained generator $G$ to the Himawari AOD products, using $H_{LR}$ and $e_h$ as inputs to obtain high-resolution Himawari AOD $H'_{HR}$. In addition, the good spatial coverage of sampled data are essential to retaining local spatial variabilities in spatial interpolation. In addition, we adopted an active window selection strategy to ensure the sampled data on a reasonable spatial scale, which will be described in detail in Section 2.3.

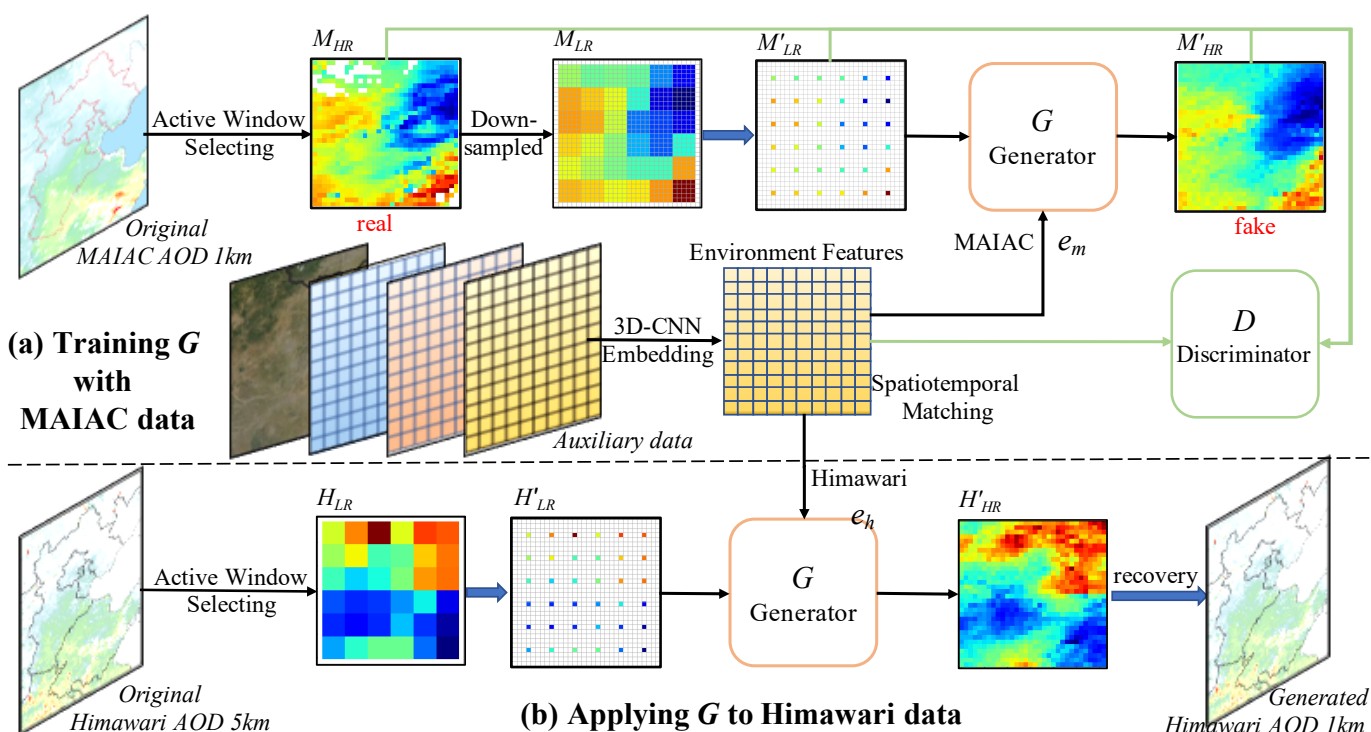

**Figure 1.** Flowchart of the proposed method, including data preprocessing, model training, and applying. Stage (**a**) is training generator $G$ with MAIAC AOD data; Stage (**b**) is applying the trained generator $G$ to Himawari AOD data. The 3D-CNN is used to extract environment features from auxiliary data. Spatiotemporal matching is required when using the environment features in the model.

**Table 1.** The notations in the following sections and Figure 1.

| Notation | Explanation |
|---|---|
| $M_{HR}$ | original high-resolution MAIAC AOD (1 km, daily) |
| $M_{LR}$ | low-resolution MAIAC AOD down-sampled from the $M_{HR}$ (5 km, daily) |
| $M_{LR}^{'}$ | sample data from $M_{LR}$ |
| $M_{HR}^{'}$ | generated high-resolution MAIAC AOD (1 km, daily) |
| $H_{LR}$ | original low-resolution Himawari AOD (5 km, hourly) |
| $H_{LR}^{'}$ | sample data from $H_{LR}$ |
| $H_{HR}$ | generated high-resolution Himawari AOD (1 km, hourly) |
| $e_m$ | environment features matching MAIAC |
| $e_h$ | environment features matching Himawari |

*2.2. Network Architecture of AeroCGAN*

Figure 2 displays the network architecture of the proposed AeroCGAN, which is based on conditional generative adversarial network (CGAN). For clarity, we will begin with a brief introduction to the concepts of GAN and CGAN. Then, based on the gaps between our research and the general conditional generation tasks, we illustrate our strategy to reform general CGAN and construct the proposed model.

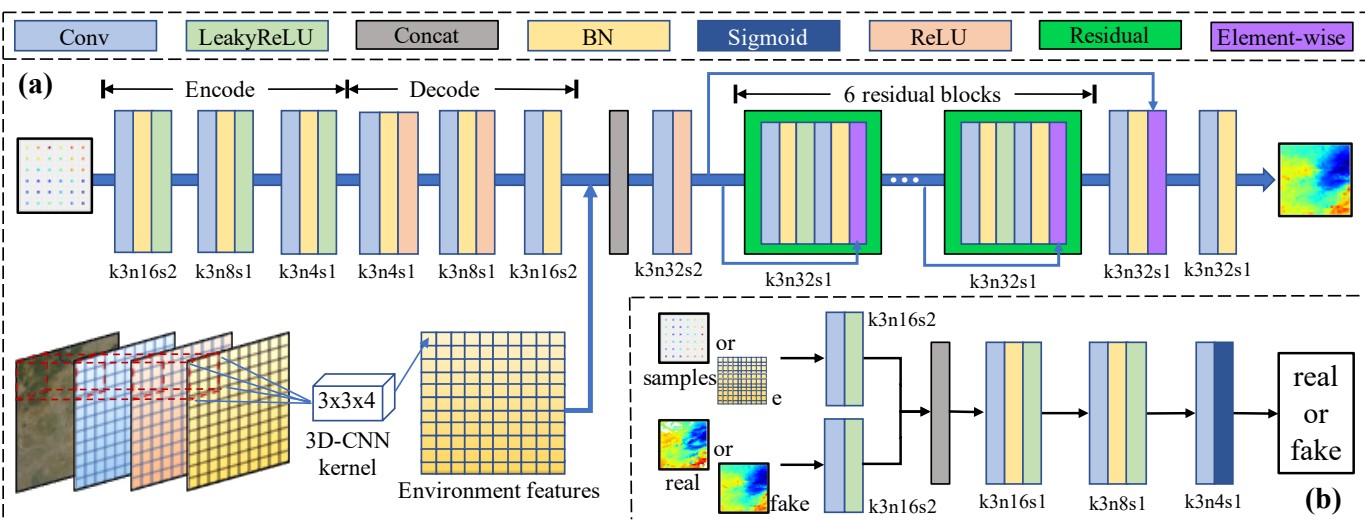

**Figure 2.** The network architecture of the proposed model with the corresponding kernel size (k), number of feature maps (n), and stride (s) indicated for each convolutional layer. (**a,b**) are the network architecture of the generator and discriminator, respectively.

The GAN introduced by Goodfellow et al. [31] contains two parts: a generator *G* that attempts to capture the data distribution and a discriminator *D* to judge whether a sample comes from the real dataset or from fake data of *G*. Usually, *G* maps a noise vector $\mathbf{z}$ from the prior distribution $p_{\mathbf{z}}(\mathbf{z})$ to the data space as $G(\mathbf{z})$, in this way to learn a generator distribution similar to the distribution $p_{data}(\mathbf{x})$ of a dataset $\mathbf{x}$. The discriminator *D* outputs a single scalar representing the probability of judging the sample $\mathbf{x}$, which comes from the real dataset rather than the generated samples of *G*. From an adversarial perspective, the *G* and *D* are similar to a two-player minimax game. The goal is to minimize $\log(1 - D(G(\mathbf{z})))$ and maximize $\log(D(\mathbf{x})) + \log(1 - D(G(\mathbf{z})))$ at the same time. During this game, the parameters $\theta_g$ of *G* are adjusted to confuse the discriminator maximally, and the parameters $\theta_d$ of *D* are adjusted to make the best judgement. The objective function of the minimax game is given as

$$\min_{\theta_g} \max_{\theta_d} \left( \mathbf{E}_{\mathbf{x} \sim p_{\text{data}}(\mathbf{x})} [\log D(\mathbf{x})] + \mathbf{E}_{\mathbf{z} \sim p_z(\mathbf{z})} [\log(1 - D(G(\mathbf{z})))] \right). \tag{1}$$

Adding the same auxiliary information **y** as a condition to the $G$ and $D$, the GAN will be extended to a conditional version named CGAN [32], which can restrict $G$ in its generation process and $D$ in its discrimination process. In general studies, the $G$ generated different random fake data on the same condition from the noise vector **z** and the condition **y**. However, the discriminator $D$ receives **x** (or $G(\mathbf{z}, \mathbf{y})$) and **y** as inputs to make a judgment based only on **y** without considering **z**. The objective function of a CGAN is formalized as follows:

$$\min_{\theta_g} \max_{\theta_d} \Big( \mathbf{E}_{\mathbf{x} \sim p_{\text{data}}(\mathbf{x})}[\log D(\mathbf{x}, \mathbf{y})] + \mathbf{E}_{\mathbf{z} \sim p_{\mathbf{z}}(\mathbf{z})}[\log(1 - D(G(\mathbf{z}, \mathbf{y}), \mathbf{y}))] \Big). \tag{2}$$

### 2.2.1. Conditions of Proposed AeroCGAN

In recent years, researchers have designed various CGAN-based networks to achieve the expected generation effect in the field of geosciences [33,34]. In this study, the proposed approach is also based on classical CGAN. We think the reasonable design of the network condition parameters will help to learn spatially deep features. Thus, according to the features of AOD data and the correlation between auxiliary data (e.g., meteorological, land-related data) and AOD, the proposed model AeroCGAN constructs two conditions: the sampled data (such as $M'_{LR}$ and $H'_{LR}$ in Figure 1) as a spatial condition for generating reasonable spatial distribution; the environment features are extracted from auxiliary data as an environmental condition for improving the accuracy and producing more realistic details.

On the one hand, we analyze characteristics of the aerosol spatial distribution. AOD data are similar to a grayscale image. It has its physical meaning, which is different from multi-spectral or hyper-spectral images. In certain geographical space, at time $t$, we can think that the AOD with a high spatial resolution can be expressed by the summary of low spatial resolution value and a difference value within the corresponding pixel range. The specific formula can be expressed as follows:

$$H^i_t = L_t + \varepsilon^i_t, \tag{3}$$

where $H^i_t$ represents the high spatial resolution AOD value, and $\varepsilon^i_t$ represents the difference value of the corresponding low spatial resolution AOD value $L_t$. There are various factors that cause the difference value $\varepsilon^i_t$, such as reflectance conditions and cloud coverage. The spatial resolution of AOD is usually measured in kilometers. Considering the spatial resolution of 1 km (MAIAC AOD) vs. 5 km (Himawari AOD), the overall different characteristics of the surface objects and the atmosphere are relatively small. In the corresponding pixel, there is a big gap between $L_t$ and $\varepsilon^i_t$ in magnitude, but with a much smaller value of $\varepsilon^i_t$. Therefore, in a certain geographical space, the high spatial resolution AOD and the low spatial resolution AOD actually contain similar spatial distribution trend features.

On the other hand, the aerosol retrieval theory is also worth mentioning. In the basic strategy of aerosol remote sensing [35], the top of atmosphere reflectance $\rho_{TOA}$ can be expressed as

$$\rho_{\text{TOA}}(\mu_s, \mu_v, \phi) = \rho_0(\mu_s, \mu_v, \phi) + \frac{T(\mu_s)T(\mu_v)\rho_s(\mu_s, \mu_v, \phi)}{[1 - \rho_s(\mu_s, \mu_v, \phi)S]}, \tag{4}$$

where $\rho_0$ represents the atmosphere path reflectance, $T$ is the transmission function describing the atmospheric effect on upward and downward reflectance, $S$ is the atmosphere backscattering ratio, and $\rho_s$ is the angular surface reflectance. These parameters ($\rho_0$, $T$, and $S$) are functions of solar zenith angle, satellite zenith angle, and solar/satellite relative azimuth angle ($\mu_s, \mu_v, \phi$). Except for the surface reflectance, each term on the right-hand side of Equation (4) is a function of the aerosol type and AOD.

According to the basic strategy of aerosol remote sensing and Equation (4), we could think that $\varepsilon^i_t$ has a strong relation with surface reflectance. Moreover, many studies illustrated the significant influence of the aerosol–radiation interaction on meteorological

forecasts, and some weather forecast centers are conducting research to facilitate the inclusion of more complex aerosol information in operational numerical weather prediction models [36,37]. Therefore, we think that aerosol and meteorological variables have interaction and cause $\varepsilon_t^i$.

Thus, according to the spatial distribution features of AOD data, it is more likely a spatial interpolation process during the encoder–decoder stage. The input of G is no longer a random noise vector $\mathbf{z}$, but a spatially sampled (and hence, low spatial resolution) data $(M_{LR}'$ or $H_{LR}')$ denoted as $\mathbf{s}$. The overall spatial distribution characteristics are still retained in $\mathbf{s}$. Thus, $\mathbf{s}$ is designed as a spatial condition. Single input data are insufficient to recover AOD with full spatial resolution. Comprehensively considering the characteristics of AOD data and its relationship with other auxiliary data (e.g., meteorological, land-related data), we used a three-dimensional convolution kernel to extract environment features from the auxiliary data and construct another environment condition $\mathbf{e}$ ($e_m$ or $e_h$). During the residual network stage, it is conditioned on $\mathbf{e}$ to correct errors and generates more realistic details. The object function of AeroCGAN is formalized as follows:

$$\min_{\theta_g} \max_{\theta_d} \Big( \mathbf{E}_{\mathbf{x} \sim p_x(\mathbf{x})}[\log D(\mathbf{x}, (\mathbf{s}, \mathbf{e}))] + \mathbf{E}_{\mathbf{s} \sim p_s(\mathbf{s})}[\log(1 - D(G(\mathbf{s}, (\mathbf{s}, \mathbf{e})), (\mathbf{s}, \mathbf{e})))] \Big). \quad (5)$$

### 2.2.2. Generator

Figure 2a illustrates the details of the architecture of generator $G$. It mainly contains three parts: an encoder–decoder network for preliminary resolution enhancement, a 3D-CNN network for environment features extraction, and a residual network for improving the accuracy. The fully convolutional encoder–decoder structure is used as the encoder which contains three two-dimensional convolution layers. Three two-dimensional transposed convolution layers are used as the decoder. Each layer of the encoder performs a zero-padding convolution with the given convolving kernel and stride length. Each layer of the decoder implements the up-sampling of the feature maps through a fractionally stride transposed convolution with the same settings as the corresponding layer of the encoder layers. In the output layers of this network, we use the Tanh activation functions. The 3D-CNN network is mainly used to extract environment features from the auxiliary data by a three-dimensional convolution kernel with a size of $3 \times 3 \times 4$. The residual network structure contains six residual blocks. Each residual block with the same layout contains two convolutional layers with a small filter kernel size of $3 \times 3$ followed by batch normalization (BN) layers and a rectified linear unit (ReLU) as the activation function.

### 2.2.3. Discriminator

Figure 2b illustrates the details of the architecture of discriminator $D$. It is a CNN similar to typical models of image classification except that we use a concatenation to merge the sampled data, environment data, and full-size real/fake data as the input. Each layer of $D$ performs a zero-padding convolution with the same settings of the encoder layers in $G$. The output of $D$ is a scalar, which indicates the input full-size image is estimated (fake) data or real data. Batch normalization is applied to all layers except for the input/output layer of $D$. This can decrease model instability and help gradients flow in the networks. The LeakyReLU activation is used after convolutions, and the ReLU activation is used after transposed convolutions. For the output layers in $D$, we use the Sigmoid activation functions.

### 2.2.4. Loss Functions

To utilize more information in the proposed AeroCGAN architecture, the conditional adversarial loss contains a spatial adversarial loss $\mathrm{Loss}_{adv1}$ and an environment adversarial loss $\mathrm{Loss}_{adv2}$. They are formulated as

$$\mathrm{Loss}_{adv1}(\theta_G, \theta_D) = -\log D\big(M_{HR}, M_{LR}'\big) - \log\big(1 - D\big(G\big(M_{LR}'\big), M_{LR}'\big)\big), \quad (6)$$

$$\mathrm{Loss}_{adv2}(\theta_G, \theta_D) = -\log D\big(M_{HR}, e_m\big) - \log\big(1 - D\big(G\big(M_{LR}'\big), e_m\big)\big). \quad (7)$$

Motivated by a previous study [38], we add a robust spatial content loss function, $\text{Loss}_{SC}(\theta_G)$, to enforce the generator $G$ to generate a high resolution AOD data similar to original AOD given as

$$\text{Loss}_{SC}(\theta_G) = \arg\min_{\theta_G} \sum_{i=1}^{n} \rho(M_{HR,i} - M'_{HR,i}), \qquad (8)$$

where $\theta_G$ means model parameters in G, and $\rho(x) = \sqrt{x^2 + \varepsilon^2}$ is the Charbonnier penalty function. $M_{HR,i}$ and $M'_{HR,i}$ refer to the real and estimated (fake) AOD, respectively. Thus, the total loss of the generator is given by

$$\text{Loss}(\theta_G, \theta_D) = \text{Loss}_{SC}(\theta_G) + \lambda_1 \text{Loss}_{adv1}(\theta_G, \theta_D) + \lambda_2 \text{Loss}_{adv2}(\theta_G, \theta_D), \qquad (9)$$

where $\lambda_1$ and $\lambda_2$ are set to balance the loss components. In this study, we alternately update $G$ and $D$, and ultimately use the optimized generator to estimate the high resolution AOD data.

### 2.3. Study Area and Data Description

Figure 3 displays the MAIAC AOD observation and the study area status. Figure 3a presents the frequency of valid MAIAC AOD observation in 2017. Figure 3b shows the annual mean MAIAC AOD observation in 2017. The red rectangle marks the study domain ranging from 113°E to 121°E and 34°N to 43°N, and Figure 3c displays the land cover of the study area. The overlaid dots represent the selected AERONET sites. As can be seen, the valid observed satellite data of this area can provide rich training data, where aerosol pollutions have high concentrations and complex properties [39,40]. Good spatial coverage of sampled data are essential to retain local spatial variabilities in spatial interpolation. On the contrary, a lower sampling density would cause a worse interpolation result. Therefore, this study collected the original data in 2017 of this area. Table 2 lists the data summary information used in this study.

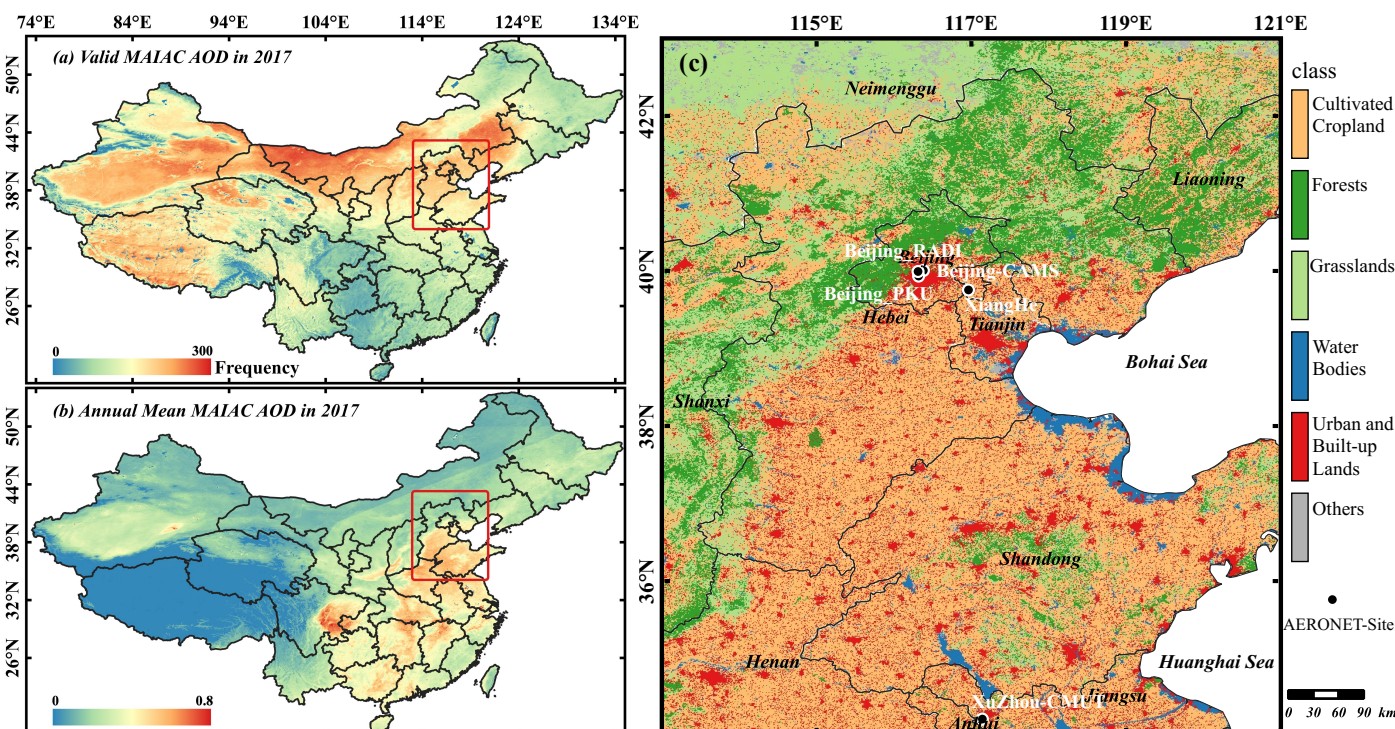

**Figure 3.** (**a**) The frequency of valid MAIAC AOD (470 nm) observation in 2017; (**b**) the annual mean MAIAC AOD (470 nm) observation in 2017. The red rectangular area is the study area; (**c**) the land cover of the study area. The five overlaid dots represent the AERONET sites.

**Table 2.** Summary of the data sources used in this study.

| Type | Variable | Resolution | | Source |
|---|---|---|---|---|
| AOD | MAIAC AOD | 1 km × 1 km | daily | MCD19A2 |
| | Himawari AOD | 5 km × 1 km | hourly | JAXA |
| | AERONET AOD | situ | hourly | AERONET |
| meteorological | 2 m air temperature | 0.1° × 0.1° | hourly | ECMWF ERA5 |
| | 10 m u-component of wind | 0.1° × 0.1° | hourly | |
| | 10 m v-component of wind | 0.1° × 0.1° | hourly | |
| | relative humidity | 0.25° × 0.25° | hourly | |
| land-related | surface reflectance | 1 km × 1 km | daily | MCD19A1 |

- MODIS MAIAC AOD products: The MODIS Multiple Angle Implication of Atmospheric Correction (MAIAC) algorithm enables simultaneous retrieval of aerosol loading at high resolution of 1 km, providing an excellent opportunity for aerosol research at finer spatial scales. It is widely used in various aerosol-related studies. This study collected the MAIAC AOD products from (MCD19A2: https://lpdaac.usgs.gov/products/mcd19a2v006/ (accessed on 15 June 2020)), and adopted it to training the initial model.
- Himawari AOD products: Himawari-8 is a Japanese geostationary satellite operated by Japan meteorology agency, carrying Advanced Himawari Imager (AHI), a multiwavelength imager [6]. The full disk observation with high temporal resolution (10 min) exhibits a prominent advantage in monitoring aerosols over the East Asia region. The AHI has 16 channels from 460 to 13,300 nm to capture visible and infrared spectral data. This study collected L3ARP Hourly Himawari AOD (L3ARP : https://www.eorc.jaxa.jp/ptree/index.html (accessed on 15 June 2020)), which is in the band of 500 nm. The trained model is transferred to Himawari data to obtain the estimation of 1-km-resolution, hourly Himawari AOD products.
- AERONET AOD data: The AERONET project is a federation of ground-based remote sensing aerosol networks. It has provided long-term, continuous, and readily accessible public domain database of aerosol optical, microphysical, and radiative properties for aerosol research and characterization, validation of satellite retrievals, and synergism with other databases. The provided spectral AOD measurements with a high temporal resolution (15 min) in the bands of 340–1060 nm, and the processing algorithms have evolved from Version 1.0 to Version 2.0 and now Version 3.0 are available from the AERONET website (https://aeronet.gsfc.nasa.gov/ (accessed on 15 June 2020)). This study collected five AERONET sites' measurements marked in Figure 3c, which is in the band of 500 nm for validation with Himawari AOD;
- Auxiliary data: The auxiliary data include surface reflectance, temperature, wind speed, and relative humidity (Table 2). The surface reflectance (SR) data are collected from (MCD19A1: https://lpdaac.usgs.gov/products/mcd19a1v006/ (accessed on 15 June 2020)). The temperature (TEM) and relative humidity (RH) are collected from the European Center for Medium-Range Weather Forecasts (ECMWF) atmospheric reanalysis products (ERA5: https://www.ecmwf.int/en/forecasts/datasets/ (accessed on 15 June 2020)), wind speed (WS) is calculated from the two wind components (10 m u-component and 10 m v-component of wind collected from ECMWF ERA5) by using the vector synthesis method. All of the auxiliary data (SR, TEM, RH, and WS) will be preprocessed and extracted as environment features by a 3D-CNN network.

Due to clouds and high reflectance conditions, satellite-retrieved AOD products usually have large percentages of missing values. Under conditions of thin cloud cover, the missing data area could be filled by the surrounding valid observation; thus, we could obtain reliable sample data for interpolation. On the contrary, when cloud coverage causes missing data in large areas, and the spatial continuity of sampled data are significantly inadequate, we could not reliably fill in missing values in such areas. Therefore, we adopted

an active window selection strategy to preprocess data. We defined an active window with a size of 32 km × 32 km. In order to ensure the quality of training data when preprocessing MAIAC AOD products, we select the areas where valid pixels are exceeding 95% in the active window, and then used a simple spatial interpolation method to complete the data block. Similarly, when estimating Himawari AOD products, we select the areas where valid pixels are exceeding 80% in the active window, and the missing data are filled by the surrounding valid observation. In this way, our active window selection strategy can select the areas where the valid pixels mostly surround the active window block for training and estimating.

After preprocessing, original MAIAC AOD is divided into single-channel AOD tiles $M_{HR}$ (1 × 32 × 32) with no repetition; then, the corresponding $M'_{LR}$ is obtained. Each AOD block covers a 0.32° × 0.32° geographic tile. We first transform these blocks linearly into float tensor images. Then, we normalize the tensor images (0.5 mean and 0.5 standard deviation) to improve training efficiency. All AOD data are mapped back to their original values in the reported accuracies. Similarly, preprocessing was used for Himawari AOD products.

### 2.4. Model Parameters and Experiment Design

Based on experiences in previous studies and our experiments, we set the slope of the leak to be 0.2 for layers with LeakyReLU activation [41]. In addition, we use the Adam optimizer [42], where $\beta_1 = 0.6$ and $\beta_2 = 0.999$, and the learning rate $\alpha$ for backpropagation is set to 0.0002. The parameter $\varepsilon$ in Equation (8) is set to $10^{-3}$. Based on our experiments and previous study [43], the parameters $\lambda_1$ and $\lambda_2$ in Equation (9) are set to $10^{-4}$ to balance the loss components. All gradients are computed using Equations (5)–(9). The multi-input neural network structure and layer sizes of generator $G$ and discriminator $D$ are shown in Figure 2. In order to demonstrate the feasibility and efficiency of the proposed model, the resulting images are evaluated by Peak Signal-to-Noise Ratio (PSNR, ranges: 0–1) and Structural SIMilarity (SSIM, ranges: 0–100 dB). Furthermore, according to the features of AOD data, we also added the RMSE index for measurement. Meanwhile, considering that the Kriging [15,44] method is widely used in geospatial interpolation, and super-resolution CNN (SRCNN) [45] is a classic super-resolution reconstruction method using deep convolutional network, we choose them for model comparison.

According to the framework shown in Figure 1, our comparative experiment mainly contains two parts:

1.  Simulation experiments with MODIS MAIAC AOD, which is corresponding to stage (a) in Figure 1. During this experiment, we acquire about 66,396 MAIAC AOD blocks in total after data preprocessing, in which 53,000 blocks are taken as the training dataset and 13,396 blocks are taken as the validation dataset. In addition, the MAIAC AOD product on 1 June 2017 was selected as testing data for displaying the performance of the proposed model.
2.  Apply the model to real Himawari AOD, which is corresponding to stage (b) in Figure 1. During this experiment, we compared the spatial distribution of the original 5 km Himawari AOD and generated 1 km Himawari AOD, which picked hourly data on 1 June 2017 for display. Moreover, we validate generated AOD with ground AERONET monitoring station data.

## 3. Results and Discussion

### 3.1. Training and Validation Model with MAIAC AOD

Table 3 displays the performance of different methods with three evaluation indices. The proposed model shows better performance in all indices, especially the method with valuable auxiliary data. In the overall spatial distribution of the AOD, the SSIM value shows that those methods have similar estimation performance of spatial distribution. In some details of resolution enhancement of the AOD, the PSNR and RMSE values demonstrate the accuracy difference. The proposed AeroCGAN model with a 3D-CNN

extracting environment features from auxiliary data achieved the best indicator values (RMSE: 0.021, PSNR: 33.24, SSIM: 0.883), and it was significantly improved compared to the other methods. In particular, the considerable RMSE improvement in AeroCGAN(3D-CNN embedding) proved that directly adding auxiliary data has limited improvement in model efficiency, but the 3D-CNN can work better by extracting high-related features from the auxiliary data.

In order to further show the detailed performance of the proposed model, we use MAIAC AOD data on 1 June 2017, in order to validate the performance of the method. The comparison results are shown in Figure 4. Overall, Figure 4a–e shows that all methods can estimate high-resolution data and fill missing data. This performance is also in line with the SSIM indicator value of Table 1. In Figure 4f–o, we pick up two experiment areas for further comparisons, and the ellipse area marks some obvious details. As can be seen, the estimated result of the classical interpolation method (Kriging) has a similar trend of the overall spatial distribution with real AOD, but it has distortion in small details. Similar performances can also be observed on SRCNN results. In other words, when improving the AOD resolution, the classical interpolation method like Kriging is prone to excessive smoothing, while the SRCNN method mainly sharpens the edges. They do not provide enough credible information for the estimation.

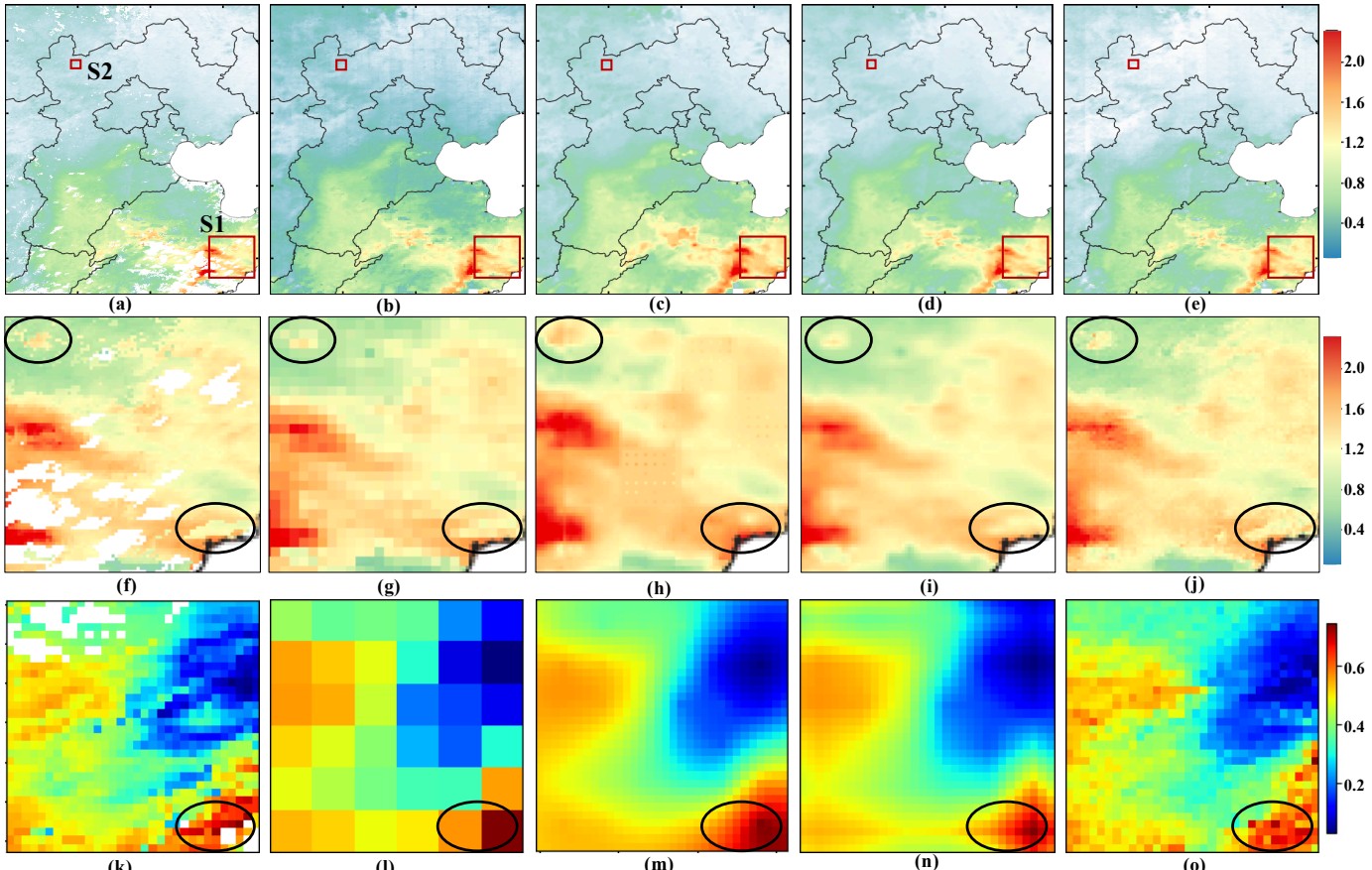

**Figure 4.** The spatial distribution performance of different method estimations with MAIAC AOD (1 June 2017). (**a**) original MAIAC AOD 1 km; (**b**) down-sampled MAIAC AOD 5 km; (**c**) Kriging estimates AOD 1 km; (**d**) SRCNN estimates AOD 1 km; (**e**) AeroCGAN estimates AOD 1 km; (**a**–**e**) are overall spatial distribution; (**f**–**j**) are the detailed comparison of big rectangular area S2; and (**k**–**o**) are the detailed comparison of small rectangular area S1. Some obvious details are marked by the ellipses area.

**Table 3.** Performances of different methods with three evaluation indices (using MODIS MAIAC AOD).

| Model | RMSE | PSNR | SSIM |
|---|---|---|---|
| Kriging | 0.036 | 28.89 | 0.868 |
| SRCNN | 0.071 | 22.81 | 0.796 |
| AeroCGAN (meteorological data) | 0.031 | 30.17 | 0.864 |
| AeroCGAN (surface reflectance) | 0.029 | 30.75 | 0.861 |
| AeroCGAN (3D-CNN embedding) | 0.021 | 33.24 | 0.883 |

In contrast, the proposed model fills missing data with a reliable value as much as possible while minimizing distortion in small details. We attribute these advantages to the architecture conditioned by auxiliary data, which corrected errors in the encoder–decoder part and generated more realistic details. Therefore, the RMSE values of Table 3 were further validated in Figure 4, and the results of the proposed model have a better correlation with the original value.

### 3.2. Applying the Model to Himawari AOD

The above result demonstrated the competitive performance of our proposed AeroC-GAN on MODIS MAIAC AOD. In addition, we applied the trained generator *G* to the Himawari data for estimating high spatiotemporal AOD. Considering that there are no real high spatiotemporal remote sensing AOD to validate our estimated results, we first compared the spatial distribution of generated Himawari 1 km AOD with original Himawari 5 km AOD, which is shown in Figure 5. In terms of the aerosol spatial distribution from 8:00 a.m. to 5:00 p.m., the generated 1 km AOD retains the spatial distribution features consistent with the original Himawari 5 km AOD. In terms of the aerosol spatial coverage, it is obviously improved in generated 1 km AOD. For example, at 8:00 a.m., the areas at the border of Liaoning and Neimenggu (red rectangular area), and north of Henan (dark rectangular area), the original Himawari observation data are sparse, and there are a lot of missing values. The corresponding generated 1 km AOD shows that the proposed model improves spatial coverage and generates high spatial resolution, which is all on a reasonable spatial scale. We attribute these advantages to the active window selection strategy. In this strategy, these kinds of missing values filled by our model are surrounding the valid observation. Similarly, this performance could also be seen at other times, which proved that our approach is robust in filling missing values. There is a truth that the recovered result tends to be unreliable if the missing data are in large areas and the spatial continuity of sampled data are significantly inadequate, which both our approach and others cannot avoid. However, our filling strategy has always been kept within a reasonable scale in a large area of missing data. For example, the central Shandong region has extensive missing data at 5:00 p.m. Although there still exists some missing value area, the region we filled is always kept in the valid observation area.

Furthermore, we use the ground measurements to validate the estimation performance, and the result is shown in Figure 6. We picked up the experiment data blocks where there are AERONET sites located. Then, we generated the improved 1-km-resolution estimates of these blocks. In addition, we extracted the filled value and the ground measurement corresponding with AERONET AOD sites. AERONET AOD at 500 nm contains all the five stations daytime hourly observation in 2017. After matching with Himawari valid AOD, we obtained 6169 records for correlation analysis. The resulting scatter plots are shown in Figure 6: (a) is the valuation of original Himawari 5 km AOD against AERONET AOD (R: 0.836, RMSE: 0.465), while (b) is the valuation of improved 1 km resolution estimated Himawari AOD against AERONET AOD (R:0.865, RMSE:0.314). Based on these values of R and RMSE, it has been proved that our estimated results exhibit higher levels of correlation with the ground AERONET monitoring station AOD data, and the biases were reduced. Moreover, the scatter dots also show that the estimated data significantly reduce the high estimation values (>4.0).

The results shown in Figures 5 and 6 all validated that the pre-trained model with MA-IAC AOD can be effectively applied to Himawari data. The generated, high spatio-temporal resolution AOD products show reasonable features in both temporal and spatial variations.

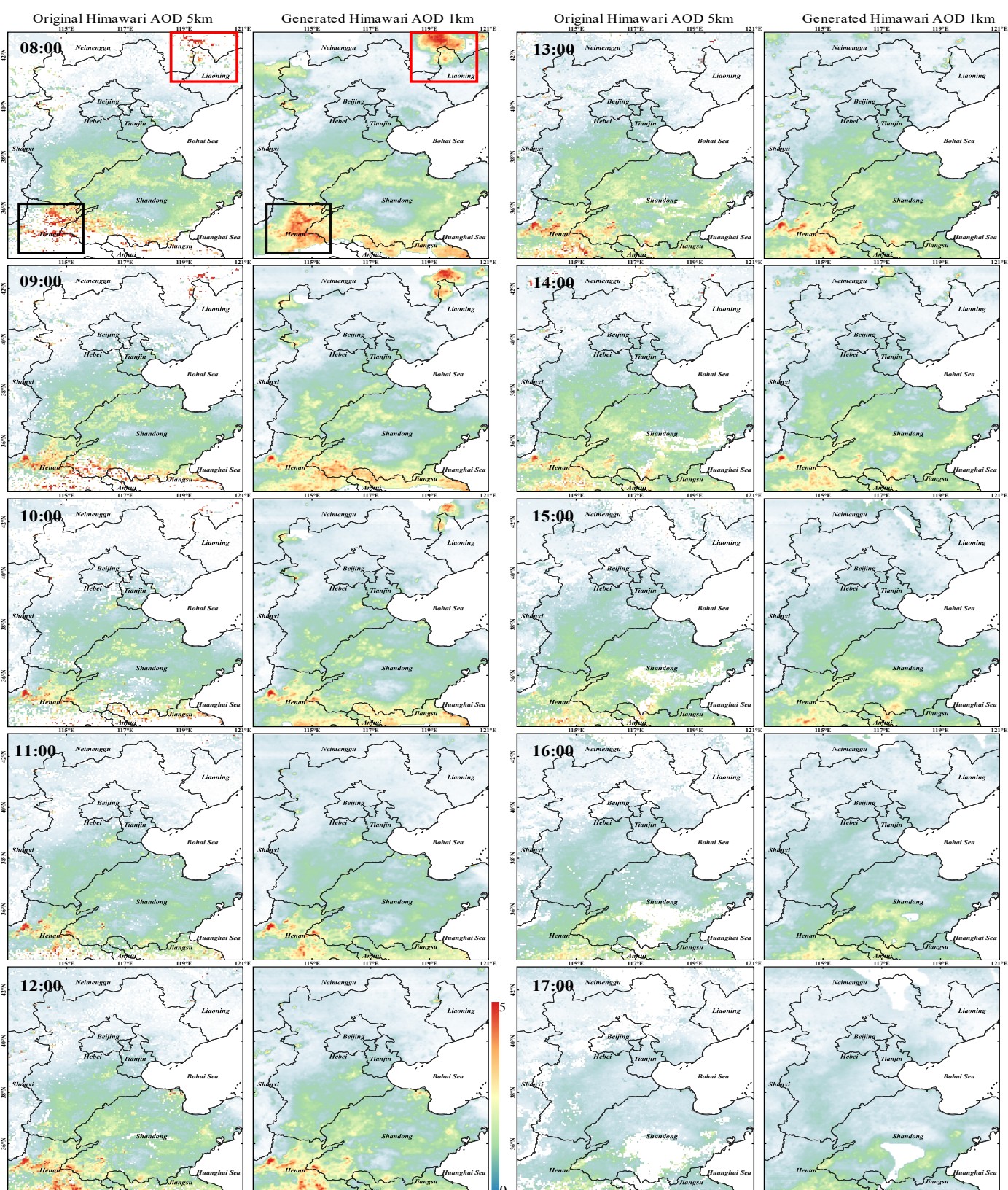

**Figure 5.** The spatial distribution comparison performance of the original Himawari 5 km AOD and generated Himawari 1 km AOD (1 June 2017, 8:00 a.m. to 5:00 p.m. at Beijing Time).

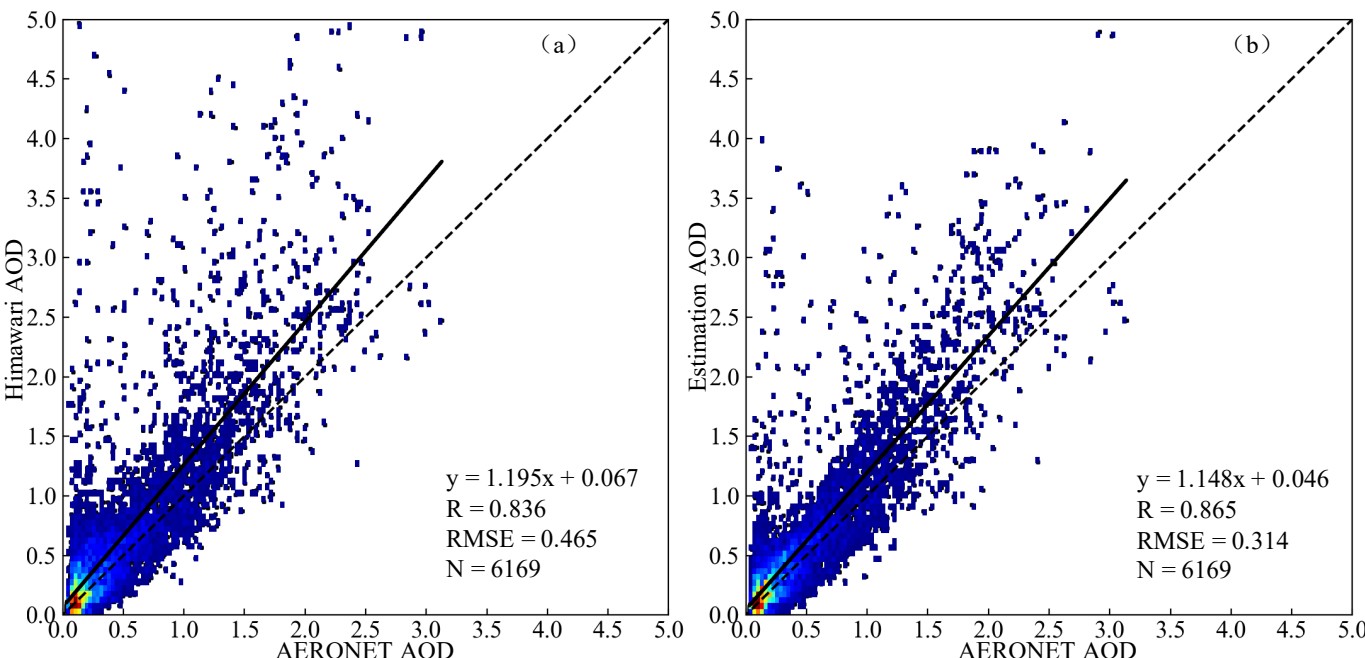

**Figure 6.** Correlation analysis of all the five AERONET sites daytime hourly observation in 2017. (**a**) the original Himawari 5 km AOD against AERONET AOD; (**b**) the estimated Himawari 1 km AOD against AERONET AOD. AOD at 500 nm.

## 4. Conclusions

In this study, we proposed an architecture and training strategy based on conditional generative adversarial networks for estimating high spatiotemporal resolution AOD (AeroCGAN). This approach achieved competitive performance by taking advantage of the high spatial resolution of MODIS MAIAC AOD products and the high temporal resolution of Himawari AOD products. It adopts MAIAC AOD to train the model and applies the trained model to Himawari data for estimating high spatiotemporal resolution AOD products. The generator adopts a conditional encoder–decoder and residual network framework for spatial estimation. We also construct a spatial and environment content loss function for correcting the process of fusion and generating more reasonable details. The visual comparison and quantitative evaluation have shown that this approach could capture deep representations of sampled spatial data and their patterns of interactions with the local environment. Meanwhile, the proposed method shows competitive performances when compared with other methods. It improves the spatial coverage and generates high spatial resolution with more realistic details, which is achieved on a reasonable spatial scale.

**Author Contributions:** Conceptualization, L.Z.; methodology, L.Z. and P.L.; software, L.Z.; validation, L.Z. and P.L.; formal analysis, L.Z.; investigation, L.Z.; resources, L.Z. and Y.Z.; data curation, L.Z. and B.S.; writing—original draft preparation, L.Z.; writing—review and editing, P.L., G.H., L.W. and J.L.; visualization, L.Z. and Y.Z.; supervision, P.L., G.H. and J.L.; project administration, P.L. and J.L.; funding acquisition, P.L., J.L. and H.Z. All authors have read and agreed to the published version of the manuscript.

**Funding:** This work was supported in part by the National Natural Science Foundation of China under Grant 41971397, Grant 61731022, Grant 81871511, and by the Henan Province Science and Technology Tackling Plan Project under Grant 212102210544.

**Conflicts of Interest:** The authors declare no conflict of interest. The funders had no role in the design of the study; in the collection, analyses, or interpretation of data; in the writing of the manuscript, or in the decision to publish the results.

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
