# Peer review of "Improved 1-km-Resolution Hourly Estimates of Aerosol Optical Depth Using Conditional Generative Adversarial Networks"

_remotesensing, doi:10.3390/rs13193834_

Round 1

Reviewer 1 Report

The paper is well written and aproaches new set of analyzes.
There some expressions that could be verified. For instance, in line 107: you use: "AODs is an atmospheric data", but it is in fact a physical parameter. And did you mean by dramatic?

In line 125: MAIAC is misspelled. And, isn't MAIAC an algorithm? In your resuts you specified it correctly.

You do not need to put AOD in plural form.

Reviewer 2 Report

A conditional generative adversarial network architecture (AeroCGAN) is proposed to generate aerosol optical depth (AOD) estimation based on the MAIAC AOD and Himawari-8 data. The resolution enhancement is attained by an encoder-decoder network, with the application of a 3D-CNN network for extracting environmental features. The resulting 1-km-resolution, hourly Himawari AOD products exhibit reasonable accuracy compared with the SKYNET data. As a whole, this article can contribute to the community by proposing a new way of merging multi-sensor AOD products. However, before judging the acceptance as a journal paper, the authors should reconsider the following issues. Also, professional editing of English and text style is mandatory, though some of the issues (but not all of them) are pointed out below.

(major)

L2 Please spell out MAIAC and 3D-CNN and add the information on the test region as well as the test period. There is a typo of “improve” at L4. “1-km-resolution hourly Himawari AOD products” should be “1-km-resolution, hourly Himawari AOD products.” A more understandable explanation would be desirable for the part of “the sampled data and environment data were designed as conditions of the generator.”

L16 “Spatio-temporal completed and accurate AOD products” – “Spatio-temporally completed and accurate AOD products.” Please add a reference after “aerosol-related studies.” For example, L. Li, High-resolution mapping of aerosol optical depth and ground aerosol coefficients for mainland China, Remote Sens. 2021, 13, 2324. https://doi.org/10.3390/rs13122324

L62 Please spell out “MAIAC” and add a brief explanation of MAIAC with a reference. Also, the authors should discuss the accuracy and reliability of the MAIAC products since this forms the basis for the present architecture for obtaining detailed AOD distributions.

L77-78 “auxiliary data (e.g., meteorological, land-related data)” – please give a complete list of the parameters that have been used as the auxiliary data in the present work. Also, in the discussion, it would be informative if the authors discuss which of the auxiliary data were most effective for interpolating the AOD distribution.

L102-120 The descriptions in these parts are not scientific enough. Please consider the following, and more concise statements should replace these parts. “The changing” should be separated into “spatial variations” and “temporal changes”. For detailed AOD studies, it is indispensable to discuss the spectral changes of AOD (e.g. Angstrom exponent). Thus, the AOD information is not necessarily a “single band image”. Please rewrite the part “The features in high- and low-resolution images are basically coherent ... and their features have the optimization observation scale” so that the reader can understand the scientific meaning clearly.

L264-265 The section “3.1 Study area and data description” should be renamed as Sec. 2.3. Also, Sec. 3.2 at L328 should be renamed as Sec. 2.4. The reason for these changes is to have “3. Results and discussion”, in accordance with the standard construction of a journal paper.

L270 Why does Fig. 3(c) show the surface elevation distribution, which does not seem to be included as the auxiliary data. Since the aerosol type is closely related to the surface coverage (e.g., urban area), would it be more meaningful to show the surface reflectance instead?

L311 “the MAIAC AOD products still have many invalid pixels” – please explain briefly how these “invalid pixels” are detected and removed.

L391-392 “These kinds of missing values filled by our model are surrounding the valid observation, so it is acceptable.” – What are precisely the “missing values” and why the accuracy and the reliability of the results are ensured? (I wonder if such missing occurs due to cloud coverage. Is this true?) More quantitative and evidence-based statements are desirable. The same applies to the following sentences, which seem to be the most important part of the evaluation of the present outcomes: “our approach is robust in filling missing values” (L395), “our filling strategy has always been kept within a reasonable scale in a large area of missing data” (L396) – the meaning of this latter sentence is not clear enough. Please consider rephrasing it.

L405-411 Please explain exactly what dataset (one year or some months, all the five AERONET stations?) has been analyzed in Fig. 6. Was there any seasonal difference or station dependence? Why Himawari (and estimated, high-resolution) AODs tend to exceed the AERONET values? Are there any reasons for the extremely high AODs?

(minor)

L19-20 “... can provide accurate globally distributed observations of spectral AODs, but limited spatial coverage[1].” – “... can provide globally distributed, accurate observations of spectral AODs, but with limited spatial coverage [1].” Please note that a space is required before “[1]”, which should be applied to all the reference numbers hereafter.

L20-23 “Model simulated AODs such as ...(WRF-Chem) [3], they could ...” – “Model-simulated AODs such as ...(WRF-Chem) [3] could ...”

L24-26 “satellite retrievals AODs ... were widely used” – “satellite retrievals of AOD ... are widely used”

L60 “an architecture(AeroCGAN)” –“an architecture (Aero-conditional generative adversarial network, AeroCGAN)”: The acronym should be spelled out here, not later at L140. Also, please insert a space before the parenthesis (which applies to all the cases hereafter).

L64 “1-km-resolution hourly estimates” – “1-km-resolution, hourly estimates”

L66-67 Please spell out “3D-CNN” and add a reference about this concept. Similarly, please add a reference to the “residual network”.

L66 “finally added” – “and finally added”

L68 “designed as conditions” – this description is too vague. Please rephrase this part.

L84 “active window selection strategy” – please explain briefly what precisely the “active window” indicates.

L93 “are also be performed” – “are also performed”

L131-132 “Table 1 listed the notations” – “Table 1 lists the notations”: please do not use the past tense to refer to the tables or figures. The same applies to all the cases such as L148, L226, L243, L266, L267, L268. L348, etc.

L132-133 “where the em and eh are environment features extracted from auxiliary data by a 3D-CNN networks.” – “where em and eh are the environment features that match MAIAC and Himawari, respectively, extracted from auxiliary data by 3D-CNN networks.” Please do not place “the” just before a symbol. The same applies to L207, L212, L223, etc.

L134-135 “... AODs data. (b) applying ... “ – “... AOD data, and (b) applying ...”: please do not omit “and”.

L138, L141 “the stage (a)” – “stage (a)”: “the” is not needed before “stage (a)” or “stage (b)”.

L142 “to obtained” – “to obtain”

L144 Please give a brief explanation of “active window selecting processing”.

L153 “contains of two parts” – “contains two parts”

L164 In principle, each equation should be a part of a sentence. Therefore, “the minimax game is as follows” should be corrected as “the minimax game is given as”, and a period must be added just after the equation. This principle applies to all the equations hereafter (L171, L224, etc.). Please use a comma after eqs.(3) and (4).

L176 “spatial deep features” – “spatially deep features”

L179 Please give a brief explanation of “the sampled data”. What is the relation with the parameters introduced in Fig. 1?

L190 “diversified” is ambiguous. Probably a better way would be something like “There are various factors the cause ..., such as ...”

L191 “Under the condition of 5 times resolution” – “Considering the spatial resolution of 1 km (MAIAC AODs) vs. 5 km (Himawari AODs)”

L193-194 “In corresponding pixel” – “In the corresponding pixel”, “and the value of \epsilon is slight” – “with a much smaller value of \epsilon”

L197-199 “is also worth to be concerned” – “is also worth mentioning”, “can be defined as:”- “can be expressed as” (Eq. (4) is not giving a definition. No colon is needed after “as”.)

L205 “formula 4” should be “Equation (4)”: The same applies at L206, L258, L333, etc.

L211 “So we can conclude” – “Therefore, we can conclude”: “So” is too colloquial.

L214-216 “The input of G is no longer a random noise vector z. But a spatially sampled data (M’_LR or H’_LR), which is denoted as s, it is low spatial resolution data.” – “The input of G is no longer a random noise vector z, but a spatially sampled (and hence, low spatial resolution) data (M’_LR or H’_LR) denoted as s.”

L217 “remained on s” – “retained in s”

L218 “Single input data is too difficult to recover AOD” – “Single input data are insufficient to recover AOD”

L220 “data(e.g. meteorological, land-related data)” – please enlist all the auxiliary data used in the present study.

L221 “to extracting”– “to extract”

L231 “layers is used”– “layers are used”

L234 “as corresponding layer” – “as the corresponding layer”

L243 In Fig. 2(b), “feak” means “fake”?

L244 “convolutional neural network” should be “CNN” since this abbreviation has already been introduced.

L248 “estimation data” – “estimated (fake) data”

L252 “output layers D” – “output layers in D”?

L256 “It is formulated as follows:” – “They are formulated as”: add a comma just after eq. (6), and a period after eq. (7).

L257 “previous study[32]” – “a previous study [32]”

L259 “(x)” of “\rho(x)” should not be a suffix. “

L257-259 “we add ... eq. (8)” – “we add a robust spatial content loss function, Loss_SC(\theta_G), to enforce ... original AOD given as”, “eq. (8)”, “where \roh(x) = ... is the Charbonnier penalty function.”

L260-261 “... is given by formula 9” – “... is given by”. No indent should be placed before “where”.

L268 “Figure 3(b) shown” – “Figure 3(b) shows”, “The red rectangular marks” – “The red square marks”

L269 Please make the “five AERONET sites” more recognizable in Fig. 3(b).

L272-273 “where aerosol pollutions are with high concentrations and complex properties” – some references are required to describe the aerosol characterization in the test region. “As good spatial coverage” – “As a good spatial coverage”

L284-286 “Himawari-8 is ... agency. It loads a multiwavelength imager called Advanced Himawari Imager (AHI), which provides full disk observation with high temporal resolution (10 min) exhibits ...” – “Himawari-8 is ... agency, carrying Advanced Himawari Imager (AHI), a multiwavelength imager [ref]. The full disk observation with high temporal resolution (10 min) exhibits ...”

L286, L291, L299, L302: the unit “\mu m” should be roman (non-italic).

L304-310 Please describe the spatial resolution of each dataset. “The surface reflectance collected from (MCD19A1” – “The surface reflectance data are collected from MCD19A1 (“

L313-318 “We adopted active window” – “We adopted an active window”, “simple spatial interpolation method” – “a simple spatial interpolation method”. Please rephrase the part “the result tends to be unreliable if ...  inadequate. We adopted activate window to ... estimating” in a more understandable manner.

L326-327 “Similarly preprocess was” – “Similar preprocessing was”

L329-332 Please give appropriate reference(s) for the leakyReLU and the Adam optimizer.

L333-335 What is “1e^-4”? “Equation (5-9)” – “Equations (5)-(9)”, “layer size” – “layer sizes”, “above Figure 2” – “in Figure 2”

L336 “result images are measured by ...” –“the resulting images are evaluated by ...”: Please spell out PSRN and SSIM, with their value ranges.

L340 The heading of this section should be “3. Results and discussion”.

L348 It is necessary to explicitly state that the results shown in Table 2 are for case (1) MODIS MAIAC AODs. In Table 2, please insert a space before each parenthesis as “AeroCGAN (meteorological data)”

L329-339 In addition, explanations (with references) of Kriging interpolation and super-resolution CNN should be given in this paragraph before showing the resulting indices in Table 2.

L369 “the classical interpolate method like Krige” – “the classical interpolating method like Kriging”

L384 In the caption of Fig. 4, “The (a) to (e)” – “Panels (a)-(e)”; the same applies to “The (f) to (j)” and “the (h) to (o)”. The region S1 is shown in (a), but where is the region S2?

L384-386 “remain the spatial distribution features” – “retain the spatial distribution features”. Please avoid repeating “spatial coverage” in a sentence. “improved on” – “improved in”

L387 “at the border of Liaoning and Neimenggu” -these region names are too small in Fig. 5. Please enlarge them.

Reviewer 3 Report

The work is devoted to a significant practical problem of the improvement in aerosol optical depth (AOD) spatial resolution. In this manuscript, generated 1-km Himawari data were retrieved on the basis of the original data with 5-km resolution. To achieve the result, authors created a network which was trained on the MODIS MAIAC AOD data with high spatial (but low temporal) resolution. Reliability of the obtained results is demonstrated.

In my opinion, methodology of the study is described in detail and correctly from the scientific point of view. Data processing was performed successively and at a high level. The presented results are new and interesting, and the conclusions are completely based on the provided analysis of the improvement in AOD spatial resolution. At this, the current text is not free from different technical errors, requiring an editorial work. I attempted to specify in the attached file both typical and most noticeable corrections.

Round 2

Reviewer 2 Report

The authors have made significant improvements to the manuscript. Following is a list of corrections/modifications that should be addressed carefully before preparing the final version.

L6-7 “and obtained the estimation of 1-km-resolution, hourly estimates Himawari AOD products” – “and obtained the estimation of 1-km-resolution, hourly Himawari AOD products”. (“estimation” and “estimates” are duplicated.)

L11-12 “The spatial distribution feature comparison and quantitative evaluation have shown” – “The spatial distribution feature comparison and quantitative evaluation over an area of Beijing and Xianghe during the year 2017 have shown” (It would be better to include the area and temporal information briefly.)

L10 “model simulates” – “model simulations” (“simulate” is a verb.)

L24 “[4], they could” – “[4] could” (“Model-simulated AOD” is the subject of this sentence.)

L28 “[6] are widely used” – “[6], are widely used” (this comma corresponds to the one at “retrievals, such as.”)

L61 “this study proposed” – “this study proposes”, “an architecture (Aero-conditional generative adversarial network, AeroCGAN)” – “an architecture of Aero-conditional generative adversarial network (AeroCGAN)”

L69 “MAIAC aerosol products has been comprehensive evaluated” – ““MAIAC aerosol products have been comprehensively evaluated”

L69 “MAIAC algorithm has” – “Since MAIAC algorithm has”

L74-75 ”in Beijing and Xianghe, where are also the main study areas of this research.” – “in Beijing and Xianghe, the main study areas of this research.”

L75 “Therefore, We”- “Therefore, we”

L116-117 “The surface reflectance has slowly temporal changes and shows high spatial variations” – “The surface reflectance shows temporally slow and spatially high variations”

L124-125 “atmosphere or geographic” – “atmospheric or geographic”

L135 “features of AOD images are monotonous and poor” – “features of AOD images tend to be monotonous and poor”

L162 “which will be described in detail in Chapter 3.1” (Move this to Sec. 2.3, see below.)

L200 “AOD data similar to a grayscale image.” – “AOD data is (or are) similar to a grayscale image.”

L207 “There are various factors the cause the difference value ...” – “There are various factors that cause the difference value ...”

L229 “aerosol and meteorological have interaction” – “aerosol and meteorological variables have interaction”

L235 “are still remained in s.” – “are still retained in s.”

L237 “data(e.g.” – “data (e.g.” (a space before the parenthesis)

L242 In Fig. 2(b) “Real or fake” – “real or fake” (as indicated inside the panel)

L244 “It mainly contains three part:” – “It mainly contains three parts:”

L274 “in G, \rho(x) =” – “in G, and \rho(x) =”

L279 – 348 Please rename “3.1. Study Area and Data Description” as “2.3. Study Area and Data Description”. This suggestion is because it is awkward to include this section (even with the introduction of MAIAC and Himawari data) in “3. Results and discussion”. In principle, all the methodologies should be described in Sec. 2. Please start Sec. 3 from L349, with the subsection of “3.1 Model Parameters and Experiment Design.”

L200 In Table 2, please insert a space between the number and unit, such as “1 km”, not “1km.”

L305 “The trained model will be transferred to Himawari data and obtained ...” – “The trained model is transferred to Himawari data to obtain ...”

L318 “and relative humidity.” – “and relative humidity (Table 2).”

L327 “satellite retrieves AOD products” – “satellite-retrieved AOD products”

L330 “reliable sample data for interpolating.”– “reliable sample data for interpolation.”

L331 “... causes missing data in large areas, sampled data spatial continuous information ...” – “... causes missing data in large areas, and the spatial continuity of sampled data ...”

L334 “an active window with size of 32 x 32.” – “an active window with a size of 32 km x 32 km.”

L336, L338 “in active window” – “in the active window”

L339 “will be filled by” – “is filled by”

L340 “could select” – “can select”

L346 “All AOD are” – “All AOD data are” (The word “AOD” is not plural.)

L351 “layerswith LeakyReLU [41] activation.” – “layers with the LeakyReLU activation [41].”

L354 “the \lambda_1 and \lambda_2” – “the parameters \lambda_1 and \lambda_2”

L355 “Equation (5)-(9).” – “Equations (5)-(9).”

L370 “was retain as” – “was selected as”

L387 “proved that: directly adding ...” – “proved that directly adding ...”

L396 “classical interpolation method Kriging” – “classical interpolation method (Kriging)”

L415 “generated 1 km AOD retain ...” – “the generated 1 km AOD retains ...”

L420 “AOD show that” – “AOD shows that”

L422-423 “These kinds of missing values filled by our model are surrounding the valid observation, so it is acceptable.” – too colloquial. Please rephrase this sentence.

L425-426 “the recovered result tends to be unreliable if the significant lack of AOD spatial continuous information, which ...” – an incomplete if-clause. Please rephrase it using “spatial continuity” as above.

L438-440 “The experiment results are displayed in Figure 6, (a) is ..., (b) is ...” – “The resulting scatter plots are shown in Figure 6: (a) is ..., while (b) is ...”

L441 “resolution estimated Himawari AOD against AERONET AOD(R:0.865” – “resolution, estimated Himawari AOD against AERONET AOD (R:0.865” (a space is needed before the parenthesis.)

L442 “Based on the R and RMSE statistical result, our estimated results has a higher correlation with ...” – “Based on these values of R and RMSE, it has been proved that our estimated results exhibit higher levels of correlation with ...”

L447-449 “The generated high spatio-temporal resolution AOD are reasonable, and these AOD show better temporal and spatial distribution.” – “The generated, high spatio-temporal resolution AOD products show reasonable features in both temporal and spatial variations.”

L453 “AOD(AeroCGAN)” – “AOD (AeroCGAN)”

Round 3

Reviewer 2 Report

L23 “drived by” – “derived by”

L61 “this study proposes a conditional generative adversarial networks-based architecture (AeroCGAN) and ...” – “this study proposes an architecture of Aero-conditional generative adversarial network (AeroCGAN) and ...”:

(Since the acronym “AeroCGAN” appears here for the first time in the text body, it is natural to spell it out here exactly.)

L294 Table 2, in the line of “land-related”: please insert a space between 1 and km.

L432 “which both our approach and others cannot avoid.” – “both of which our approach and others cannot avoid.”

L444 “in Figure 6, (a) is” – “in Figure 6: (a) is”